# Do Virtual Campuses Provide Quality Education? A Study on the Perception of Higher Education

Ellián Tuero [1][ID], Lucía Álvarez-Blanco [2,*], Isabel C. Ayala-Galavis [1], Celia Galve-González [1] and Ana B. Bernardo [1][ID]

1   Department of Psychology, University of Oviedo, 33003 Oviedo, Spain
2   Department of Educational Sciences, University of Oviedo, 33005 Oviedo, Spain
*   Correspondence: alvarezblucia@uniovi.es

**Abstract:** In recent years, there has been an increase in the use of technologies in all aspects of daily life, especially in educational contexts. Indeed, in most universities, using a virtual campus as a support for teaching is now a general practice, even in face-to-face teaching. However, although there are multiple studies on the quality of education and services provided by virtual campus platforms, as well as statistics on the use of the various tools and forums, very few studies have considered students' perceptions of the quality perceived by students concerning the pedagogical use that teachers make of virtual campuses. Even fewer studies have examined this as a function of what year the students are in. In order to examine this more thoroughly, an ad hoc questionnaire was applied to 783 students enrolled in the first three years at various universities in Spain, covering all knowledge areas. The results show that first-year students had more positive opinions of the quality of their teachers' virtual-campus practices than students in subsequent years. More specifically, those first-year students perceived greater encouragement and motivation from the teaching staff and more communication between teachers and students. These findings suggest the need for good teaching practices that consider motivation, communication and collaborative groups, not only during the first year, but also throughout university courses in order to ensure quality education.

**Keywords:** higher education; virtual campus; quality; virtual education; digital environments

## 1. Introduction

The challenge facing current university teachers in the 21st century is educating skilled professionals capable of adapting to the demands and requirements of a globalized society that is in constant flux and is dominated by information and communication technologies [1–3]. In this regard, in a context in which the amount of information seems unsurmountable [4], future graduates will have to develop skills that will allow them to learn autonomously, practically, and continuously, and maintain their engagement throughout their lives [5]. More specifically, and as established by the European Higher Education Area (EHEA), competencies such as personal initiative, individual responsibility, and particularly critical thinking, digital competence, and collaborative working are essential and have become more predominant in this new era [6,7].

Against this backdrop, universities have seen the need to revamp and adapt their traditional teaching methods in order to encourage the learning of these competencies beyond the walls of the traditional classroom [8,9]. In the last decade, and in response to the European convergence process, new, more suitable spaces have been added to university teaching [10], based principally on ICT [11,12]. This is the case with virtual campuses, the use of which as a teaching support is nowadays one of the most widespread practices in these institutions [13,14]. In fact, according to González-González and Infante-Moro [15] (p. 1), they have become a key part of contemporary higher education by allowing "continued lifelong education and learning, and making collaboration, expansion, the relationship

with society, and knowledge transfer possible". All of which are in line with the new teaching–learning paradigm demanded by the EHEA.

The use that different universities make of these virtual environments is so varied—as is the terminology used within the field: web educational platforms, virtual spaces, virtual classrooms, etc.—that according to Urbina and Salinas [16] (p. 6), it is difficult to find unanimity in the scientific literature about how they are defined. Despite that, and according to these authors, there does seem to be a consensus that virtual campuses, rather than correlating to physical university campuses, are "web sites available to an educational community, with the facility to provide teaching resources, and communication and interaction functions". Nevertheless, although they were created to make the educational services provided by the universities more available—teaching materials, online library access, etc. [17]—currently, thanks to advances in educational information technology, there are many possibilities. These include a better accessibility from any internet-connected device [18,19], making the teaching–learning process possible in any context and at any time [20].

Apart from the above, virtual campuses are spaces which provide students with information relevant to their courses such as teaching guides, syllabuses and coursework exercises [21]. They also allow students more opportunity for interaction with teachers [22]. Students can consult their teachers more, exchange more messages (not only on academic topics) and receive more tutoring from their teachers by being able to manage these processes more easily and quickly given their essentially online nature [23]. In this regard, the possibilities of these more direct, varied, and educational interactions [24] in these qualification or evaluation instruments [25], make it possible for students to receive more continuous attention and allow a more individualized evaluation process, providing other means of communication and contact outside the traditional classroom [26]. Similarly, virtual campuses help students to interact more with their classmates by, for example, participation in forums and the use of interactive resources (Webquests, Blogs, Wikis, etc.) and tools in these online spaces which encourage collaborative learning [27,28].

Nonetheless, and even though there is this methodological versatility [14], researchers such as Area et al. [13] have warned of the domination of expository teaching in which the teaching role is that of a mere transmitter of knowledge. There are, however, professionals who encourage more active, autonomous, collaborative work from their students in the teaching–learning process, relegating their own roles to the background [29].

This being the case, and having confirmed that the success of these virtual spaces comes from the "multiple educational activities, from permanent accompaniment ( . . . ) to the generation of affective ties of value and respect, that is, to human interaction" [30] (p. 53), the overall trend, nowadays, for teachers' use of virtual campuses is as simple support for—and even copies of—the activities in their traditional face-to-face classes [22]. These technological spaces are flooded with notes, class presentations, and an endless duplication of materials [19], which turns them into repositories of information and content [31], consequently failing to take advantage of the interactive and communicative potential there is for the individualization of learning [19].

In accordance with these ideas, what provides quality to teaching is not the virtual campus tools in and of themselves [23,32], but rather that "they acquire pedagogical value ( . . . ) as mediating artefacts between the teacher and student or between peers which provides a unique virtual educational context facilitating interactive processes of co-construction of knowledge" [33] (p. 164). Only to the extent that the use of these virtual tools are in effect combining teaching-support-quality (in face-to-face or distance teaching), will higher educational institutions be pursuing excellence, educational efficacy [34], and consequently student motivation [35], satisfaction, and performance as indicators of the quality of their educational systems [36].

In this regard, the 2016 UNIVERSITIC report showed that universities should continue to develop, implement, and promote good practices related to virtual teaching [37]. This inevitably includes research and the systematic evaluation of the quality of the teaching

in these technological learning environments, considering not only criteria based on the technological infrastructure, teaching methodologies used, and the results of learning, but also on users' opinions, especially student users [38]. Although in recent years there has been growing interest in the scientific community about the study of the quality of teaching practices via virtual campuses in higher education [39–41], there are few studies that have examined this from the student perspective [42].

*Objectives and Hypotheses*

In light of this, and bearing in mind that new students have different needs in terms of attention, monitoring, and evaluation of the learning process compared to more experienced students [8], the general objective of this current study is to analyse students' perceptions, by university year, of the teaching practices delivered through the virtual campuses as a support to provide quality to their university teaching. A more specific objective is to attempt to determine whether there are statistically significant differences in student perceptions of student–teacher interactions encouraged by teaching practices through the virtual campus. To that end, we have formulated the following research hypotheses: (1) first-year students will ascribe more importance to teachers' teaching uses of the virtual campus as a quality complement to their classes than students in later years, (2) first-year students will ascribe more importance to teacher–student interaction-communication driven by the teaching use of virtual campus tools, and a final hypothesis that applies to the entire sample is (3) that there will be no statistically significant differences between the three academic years in students' perceptions about the teachers' promotion of motivation in interactions via the virtual campus.

## 2. Materials and Methods

In order to respond to the objectives, we performed an ex post facto study with a descriptive, inferential research design.

### 2.1. Participants

A total of 783 university students participated in the study. Three quarters were women (74.58%, *n* = 584) and one quarter were men (25.42%, *n* = 199). They were aged between 19 and 59 years old (M = 22.43, SD = 7.04). All were studying courses in various Spanish higher education institutions: the Universidad Nacional de Educación a Distancia (The National Distance Learning University)—UNED—(*n* = 112), and other universities in Spain (*n* = 31), with the vast majority studying at the University of Oviedo (*n* = 636). Most students were in the first year of their degree course (*n* = 381), followed by second-year (*n* = 256), and third-year (*n* = 146) students. The students were studying subjects that covered all knowledge areas: social and legal sciences (54.3%), engineering and architecture (3.8%), health sciences (37.2%), arts and humanities (4.1%), and science (0.6%).

Given the sociocultural and historical background of the students (i.e., the knowledge and information society characterized by the use of new technologies), 98% of the sample reported having internet-connected electronic devices with which they could access, among other things, the virtual campus during their courses of study.

### 2.2. Instruments

The data collection was via a questionnaire entitled "Analysis of university students' perceptions of virtual campuses in the European Higher Education Area". This was created ad hoc and validated in previous studies [43]. The reliability, in terms of internal consistency, was calculated using the Cronbach alpha, giving a value of 0.894, which according to O'Dwyer and Bernauer [44], is a more than acceptable value.

The initial questionnaire was made up of 44 items in eight blocks: Block A–Introduction and collection of sociodemographic data (9 items with semi-open responses); Block B–Availability of resources at home (5 items with yes/no responses); Block C–Teacher planning (4 items); Block D–Content (7 items); Block E–Methodology (5 items); Block

F–Communication (5 items); Block G–Evaluation (5 items), and Block H–Digital competence (4 items). Blocks C, D, E, F, G, and H were Likert-type responses with four options (e.g., 1 = completely disagree, 2 = disagree, 3 = agree, and 4 = completely agree).

We selected 12 items from various blocks for the current study in order to achieve our objectives, making up two dimensions of study. The 44 items assessed aspects of the teachers, students, and the virtual campus in general. Therefore, for this specific study, given its objective and hypotheses, 12 items were selected corresponding to what students might think about teachers' usage of the virtual campus, as well as the interaction that they could maintain through using this tool. The following table establishes the classification of these items with their respective dimensions (see Table 1).

**Table 1.** Dimensions and items selected to study university students' perceptions about whether teaching practices in virtual campuses provide quality to education.

| Dimensions | Item | Descriptions |
|---|---|---|
| Teachers' teaching practice in a virtual campus as support for the quality of teaching | 1. | The subject content in the virtual campus is up to date. |
| | 2. | Teachers have up to date, specialized training in managing the virtual campus. |
| | 3. | Activities are published on the virtual campus which encourage the discussion of ideas, debate, etc. |
| | 4. | Teachers ask for an evaluation of the teaching and technical content of the subject. |
| | 5. | Teachers give guidance and advice through the virtual campus. |
| | 6. | Teachers demonstrate a positive attitude towards using the virtual campus. |
| Teachers' making use of the potential of the virtual campus for interaction with students | 7. | I only get information via the virtual campus about subject grades (messaging, individual scores . . . ). |
| | 8. | Communication with teachers via the virtual campus flows well. |
| | 9. | Teachers often contact me through the virtual campus. |
| | 10. | Teachers respond satisfactorily to queries and observations. |
| | 11. | Teachers respond quickly to queries and observations. |
| | 12. | Teachers promote motivation in their interactions via the virtual campus. |

Source: researchers' own work.

### 2.3. Procedure

For this study, an incidental sampling was used, with the intention that the sample be as diverse as possible. The researchers contacted teachers in various universities with easy access who were willing to collaborate, explaining the aim of the study in detail and the feasibility of applying the instrument to their students voluntarily and anonymously. Teachers were selected with the following inclusion criteria, those who: (a) preferentially taught undergraduate degrees, and (b) used virtual campuses in their subjects.

The teachers were responsible for administering the questionnaire to their own students following the researchers' instructions, and offering the students the option to participate in the study by completing the questionnaire online in the virtual campus of the teachers' subject. Before completing the questionnaire, the students were informed of the study objectives, its confidential nature, and the time needed to complete the electronic questionnaire, which was around 30 min.

### 2.4. Data Analysis

We used the SPSS v.24. statistical software, starting with a descriptive analysis (i.e., measures of central tendency and variability). Following that, the normality of the distribution was checked in order to select the comparative analysis to perform. According to the Kolmogorov–Smirnov statistic (df > 50 and $p < 0.001$ in all cases), we confirmed the existence of differences in the samples, meaning the data were not normally distributed.

Subsequently, we carried out non-parametric tests, performing the Kruskal–Wallis analysis to determine possible differences between the groups according to their academic year (i.e., 1st, 2nd, and 3rd) via the Chi-squared statistic. Effect sizes were determined using Cohen's *d*, with values between 0.20 and 0.49 indicating small effect sizes, values between 0.50 and 0.79 indicating moderate effect sizes, and values over 0.80 indicating

large effect sizes [45]. Correlational analyses were also performed using the Pearson correlation coefficient.

## 3. Results

This section covers the two dimensions of analysis about university students' perceptions of the quality of teaching practices via the virtual campus as: (1) a complementary support resource which adds quality to the education received, and (2) a potential tool for encouraging teacher–student interaction.

As Table 2 shows, most of the students surveyed (M = 2.83; SD = 0.690) thought that the subject content in the virtual campus at their universities was up to date (73.9%), although there were some very different scores (with close scores for agree and disagree) in response to the activities being published on the virtual campus which encouraged the discussion of ideas, debate, etc., and teachers asking for an evaluation of the teaching and technical content of the subject (M = 2.36; SD = 0.767).

**Table 2.** Percentages, mean, and standard deviation for the items in Dimension 1: the campus as a complimentary support resource adding quality to the education received.

| Items | Percentage (%) | | | | M | SD |
|---|---|---|---|---|---|---|
| | CD | D | A | CA | | |
| 1. The subject content in the virtual campus is up to date. | 3.8 | 22.2 | 61 | 12.9 | 2.83 | 0.690 |
| 2. Teachers have up to date, specialized training in managing the virtual campus. | 9.1 | 30.5 | 53.9 | 6.5 | 2.58 | 0.746 |
| 3. Activities are published on the virtual campus which encourage the discussion of ideas, debate, etc. | 14.6 | 43.9 | 37.7 | 3.8 | 2.26 | 0.773 |
| 4. Teachers ask for an evaluation of the teaching and technical content of the subject. | 13.8 | 40.7 | 41.4 | 4.1 | 2.36 | 0.767 |
| 5. Teachers give guidance and advice through the virtual campus. | 15.7 | 41.8 | 38.4 | 4.1 | 2.31 | 0.781 |
| 6. Teachers demonstrate a positive attitude towards using the virtual campus. | 7.5 | 24.4 | 58 | 10.1 | 2.71 | 0.749 |

Source: researchers' own work.

In addition, referring to the teaching role, 60.4% of the students felt that the teachers had up to date, specialized training for successfully managing the virtual campus for their various subjects (M = 2.58; SD = 0.746), with a similarly high percentage of students reporting that teachers demonstrated a positive attitude towards using the virtual campus (68.1%, M = 2.71; SD = 0.749). Lastly, when asked about whether teachers gave guidance and advice through the virtual campus (M = 2.31; SD = 0.781), the majority disagreed or completely disagreed (57.5%).

The Kruskal–Wallis test was performed to determine whether there were statistically significant differences between the surveyed students depending on their academic year. The results are given in Table 3.

Significant differences were only found in one of the variables in this dimension, with first-year students more in agreement that activities are published on the virtual campus which encourage the discussion of ideas, debate, etc. ($\chi^2$ = 15.33; $p < 0.001$; $d$ = 0.264), with a small effect size. This was followed by second-year students ($n$ = 256; mean rank = 368.93), with third-year students being least in agreement that there were these types of activities in the virtual campus ($n$ = 146; mean rank = 355.09).

After reviewing the main results for the first dimension of the study, we continued with the scores for the second dimension (Table 4).

**Table 3.** Comparative analysis for Dimension 1: the campus as a complimentary support resource adding quality to the education received.

| Items | Year | $n$ | Mean Rank | $\chi^2$ | $p$ | Cohen's $d$ |
|---|---|---|---|---|---|---|
| 1. The subject content in the virtual campus is up to date. | 1st<br>2nd<br>3rd | 381<br>256<br>146 | 400.09<br>390.88<br>372.87 | 2.02 | 0.363 | |
| 2. Teachers have up to date, specialized training in managing the virtual campus. | 1st<br>2nd<br>3rd | 381<br>256<br>146 | 402.38<br>383.28<br>380.19 | 1.94 | 0.378 | |
| 3. Activities are published on the virtual campus which encourage the discussion of ideas, debate, etc. | 1st<br>2nd<br>3rd | 381<br>256<br>146 | 421.65<br>368.93<br>355.09 | 15.33 | 0.000 | 0.264 |
| 4. Teachers ask for an evaluation of the teaching and technical content of the subject. | 1st<br>2nd<br>3rd | 381<br>256<br>146 | 401.70<br>390.43<br>369.44 | 2.52 | 0.284 | |
| 5. Teachers give guidance and advice through the virtual campus. | 1st<br>2nd<br>3rd | 381<br>256<br>146 | 403.92<br>377.21<br>386.85 | 2.57 | 0.276 | |
| 6. Teachers demonstrate a positive attitude towards using the virtual campus. | 1st<br>2nd<br>3rd | 381<br>256<br>146 | 405.64<br>383.39<br>371.49 | 3.74 | 0.153 | |

Source: researchers' own work.

**Table 4.** Percentages, mean and standard deviation for the items in Dimension 2: the campus as a potential tool for teacher–student interaction.

| Items | Percentage (%) | | | | M | SD |
|---|---|---|---|---|---|---|
| | CD | D | A | CA | | |
| 7. I only get information via the virtual campus about subject grades (messaging, individual scores . . . ). | 6.1 | 11.6 | 54 | 28.2 | 3.04 | 0.802 |
| 8. Communication with teachers via the virtual campus flows well. | 19.3 | 43.7 | 31.4 | 5.6 | 2.23 | 0.823 |
| 9. Teachers often contact me through the virtual campus. | 19.4 | 41.5 | 33.1 | 6 | 2.26 | 0.837 |
| 10. Teachers respond satisfactorily to questions and observations. | 8.4 | 20.6 | 64 | 7 | 2.70 | 0.722 |
| 11. Teachers respond quickly to questions and observations. | 10.5 | 35.2 | 49.4 | 4.9 | 2.49 | 0.746 |
| 12. Teachers encourage motivation in their interactions via the virtual campus. | 20.4 | 47.9 | 27.1 | 4.6 | 2.16 | 0.797 |

Source: researchers' own work.

As the table shows, in this second dimension the students' had a negative assessment of the potential of the virtual campus as a tool or vehicle promoting communication with teachers. More than three quarters of those surveyed (82.2%) reported that the fundamental content about which they received information from the virtual campus was about subject grades (M = 3.04; SD = 0.802), rather than other non-curricular aspects that would contribute to increased student motivation and involvement. In addition, and in relation to this communication, most subjects reported that teachers responded satisfactorily to questions and observations via the virtual campus (M = 2.70; SD = 0.722).

However, there did appear to be certain gaps in the nature of this communication, which was not frequent (M = 2.26; SD = 0.837), nor did it flow well (M = 2.23; SD = 0.823). In addition, the speed with which the teachers respond to questions and observations was rapid for only 54.3% of the sample. The details of these communications helps us understand that 68.3% of students reported that teachers did not encourage motivation in their interactions (M = 2.16; SD = 0.797).

Finally, following the Kruskal–Wallis test, significant differences were found in five of the six variables analyzed (see Table 5), with small effect sizes in each case.

**Table 5.** Comparative analysis for Dimension 2: the campus as a potential tool for teacher–student interaction.

| Items | Year | n | Mean Rank | $\chi^2$ | p | Cohen's d |
|---|---|---|---|---|---|---|
| 7. I only get information via the virtual campus about subject grades (messaging, individual scores … ). | 1st<br>2nd<br>3rd | 381<br>256<br>146 | 398.57<br>374.20<br>406.07 | 3.02 | 0.221 | |
| 8. Communication with teachers via the virtual campus flows well. | 1st<br>2nd<br>3rd | 381<br>256<br>146 | 420.99<br>366.01<br>361.92 | 13.91 | 0.001 | 0.249 |
| 9. Teachers often contact me through the virtual campus. | 1st<br>2nd<br>3rd | 381<br>256<br>146 | 415.06<br>378.66<br>355.21 | 9.847 | 0.007 | 0.202 |
| 10. Teachers respond satisfactorily to questions and observations. | 1st<br>2nd<br>3rd | 381<br>256<br>146 | 407.25<br>390.43<br>354.96 | 7.769 | 0.021 | 0.173 |
| 11. Teachers respond quickly to questions and observations. | 1st<br>2nd<br>3rd | 381<br>256<br>146 | 427.76<br>371.52<br>334.59 | 25.21 | 0.000 | 0.35 |
| 12. Teachers encourage motivation in their interactions via the virtual campus. | 1st<br>2nd<br>3rd | 381<br>256<br>146 | 419.05<br>370.43<br>359.24 | 12.58 | 0.002 | 0.240 |

Source: researchers' own work.

An overall examination of the items in Table 5, focusing on those where differences were seen, shows that the first-year students had higher evaluations of the campus as a tool promoting interaction with the teachers, especially compared to the third-year students who disagreed most strongly with the idea of a communicative process driven by their respective virtual campuses. The values of the mean ranks were higher for the first-year students in variables referring to communication with teachers that was frequent (mean rank = 415.06; $\chi^2$ = 9.847; $p$ = 0.007; $d$ = 0.020); that flowed well (mean rank = 420.99; $\chi^2$ = 13.91; $p$ = 0.001; $d$ = 0.024), was satisfactory (mean rank = 407.25; $\chi^2$ = 7.769; $p$ = 0.021; $d$ = 0.017), and was a rapid response to student queries (mean rank = 427.76; $\chi^2$ = 25.21; $p$ < 0.001; $d$ = 0.035). The first-year students also perceived better encouragement of motivation in interactions by teachers via the virtual campus (mean rank = 419.05; $\chi^2$ = 12.58; $p$ = 0.002; $d$ = 0.024) than the second- or third-year students.

## 4. Discussion

Starting from the premise that a new student would need more support in their learning processes to be able to make the most of their entry into higher education, and with that, to properly adapt to the academic demands involved [46], the general objective of this study was to examine whether students' perceptions about the quality of the teaching practices delivered via virtual campuses would vary depending on which academic year they were in. In the first dimension of the analysis, we expected the first-year students to have higher ratings for teachers' teaching practices in these spaces as a supporting teaching quality (H1); however, the results from the variables in this dimension (items 1–6 in Table 3) did not support this hypothesis, although one of the variables (item 3 in Table 3) did exhibit statistically significant differences between the three years. This item referred to the didactic use teachers made of these platforms to publish activities which encouraged a discussion of ideas, debate, thought, and critical thinking, etc. This finding is in line with Bangert [47] who, in a study looking at master's students' evaluations of a nursing course, concluded that students positively evaluated the quality of activities set by teachers, via the virtual campus, promoting debate and discussion, which allowed for a better understanding of course content [48]; however, most of the students surveyed in our study exhibited the opposite opinion. This was what Area et al. [13] and Cisneros [14] found, demonstrating that generally, and regardless of the year the teaching took place in, the tendency was to use these spaces principally as mere repositories of information and, to a lesser extent,

as a teaching resource for more active learning activities needing collaboration between students (e.g., discussion, exchange of ideas, etc.).

Continuing with the second dimension, referring to teachers' making the most of the interactive and communicative potential of the virtual campus, once again it was the first-year students who perceived more encouragement of teacher–student interactions via this resource, which confirms our second hypothesis (see items 7–12 in Table 3). Considering the fact that many studies have confirmed that the transition to university produces significant academic, psychological, and social changes that can on occasion lead to insecurity and disorientation, better attention and monitoring by teachers becomes essential for better student adaptation to the university context. Aspects such as interaction, feedback, and tutorials during the first year promote better satisfaction with the teaching received. For that reason, our results show that from the perspective of the students surveyed, it was the first-year students who most felt that teachers maintained a frequent contact and flowing communication with them, and who responded quickest and most satisfactorily to their queries and observations. In this way, they understood that their teachers made better use of the interactive and communicative tools offered by the virtual campus (see items 7–11 in Table 5). This is along similar lines to the findings from Bangert [49] with master's students. That author found that a high percentage of students reported that their teachers were always accessible and not only responded rapidly to their queries, but also gave them supportive feedback, effective, personalized communication, exhibited high levels of concern that they learn, and encouraged them to perform their tasks better. However, our results contrast those found by Fariña-Vargas et al. [22] who, in a study in the virtual classrooms of La Laguna University, found that teachers who taught various classes made little use of interactive resources via the virtual campus, offering hardly any educational feedback. These results agree with those from Area et al. [13], who warned of minimal continued tutoring or teacher–student feedback. According to that study, a lack of this teaching feedback could be an incentive for students to participate less in academic activities, and consequently for a reduced interest and curiosity about their subjects [50].

Lastly, the third research hypothesis, within this same dimension, was rejected. Against our expectations, our results confirmed that there were statistically significant differences between the three academic years in the student perceptions about the encouragement of motivation by teachers in interactions via the virtual campus. Once again, it was the first-year students who exhibited the most appreciation of this encouragement compared to their second- and third-year classmates. These results contrast those found by Álvarez et al. [43], who reported in their study that 83.9% of undergraduate students surveyed, mainly in their first year, reported that their teachers did not motivate them enough in their interactions via the virtual campus. Nevertheless, one of the strategies to implement in order to achieve a standard of quality in teaching is for motivating university students via individualized attention which fosters and incentivizes their curiosity and desire to learn [51]. In order to do that, the teacher must be flexible in the face of the various challenges and concerns that students may present, and of course, ensure a friendly and harmonious working climate, encouraging participation in discussion forums through the creation of different types of debate. On similar lines, Rojas et al. [52] suggested that if good teaching practices were implemented via virtual campuses, it could notably increase student motivation, so that students would persist in their courses and be more likely to successfully complete their education.

Similarly, in research by Martínez et al. [43], it was first-year students who made the best use of the support tools and teaching materials the teachers made available on the various virtual platforms (e.g., teaching guides, syllabuses, notes, activities, etc.) and who used forums and bulletin boards more, with the latter being the space the teachers usually used to inform students about items of interest related to their subjects. In our study, on similar lines, the fact that the first-year students were the ones who demonstrated the best perceptions of teaching quality supported by the virtual tools provided by the campus may be precisely due to the fact that during the first year the teachers use them more,

incentivizing their students to use them as well. This would be in line with Feliz [53] by indicating that these tools need significant teacher involvement to obtain full use of them; therefore, if teachers do not make proper use of the virtual campus, it will also be difficult for students to do so.

This study does have some limitations. It would have been interesting to have had a greater representation of students from private or fully virtual universities, as well as the teachers' own opinions, which would have allowed the identification of where opinions agree and differ, giving guidance to future informative and training action aimed at both agents in the teaching–learning process. It would also have been interesting to explore the possible influence of gender, knowledge area, and university type on the opinions collected. For future research, it could also be interesting to compare the students in their first and fourth years to analyze the differences between them, as well as those that could belong to similar areas or disciplines. There might be differences between those new students and those who are about to graduate in their perception about the use of the campus by teachers, as well as the importance of the use of the virtual campus for the students of a 1st year course. These considerations could be included in future research which, in addition to the above, could complement this positivist methodology with another interpretive paradigm that would provide information on what the items in the questionnaire mean individually to the surveyed students. In this respect, we would suggest discussion groups as an instrument, with the information treated by a content analysis.

## 5. Conclusions

In summary, and in light of the above, it seems essential that teachers deliver appropriate practice, through the support of the virtual campus, not only during the first year but also in subsequent years. In consequence, higher education institutions, and even more so the teachers within them, should be aware of the need to adapt these practices to the profiles of the students in the different academic years so that they can deliver quality teaching, which involves offering the students what they require: functional knowledge; strong, effective communication processes; and a continual updating of the subjects of digital skills and student tutoring. Besides that, future challenges to develop the quality of virtual campuses could include activities that improve the collaborative learning between students. In addition, it would be important for teachers to consider the full potential of the virtual campus and to take advantage of all the tools it offers through training and education in order to avoid using it merely as a repository, especially for those teaching first-year students, because as this study found, new students need more involvement from their teachers through the virtual campus.

**Author Contributions:** Conceptualization, E.T., L.Á.-B., I.C.A.-G., C.G.-G. and A.B.B.; methodology, E.T. and L.Á.-B.; software, E.T. and L.Á.-B.; validation, E.T., L.Á.-B., I.C.A.-G., C.G.-G. and A.B.B.; formal analysis, E.T. and L.Á.-B.; investigation, E.T., L.Á.-B., I.C.A.-G., C.G.-G. and A.B.B.; resources, E.T., L.Á.-B. and I.C.A.-G.; data curation, E.T., L.Á.-B. and I.C.A.-G.; writing—original draft preparation, E.T., L.Á.-B., I.C.A.-G., C.G.-G. and A.B.B.; writing—review and editing, E.T., L.Á.-B., I.C.A.-G. and C.G.-G.; visualization, L.Á.-B.; supervision, E.T. and L.Á.-B.; project administration, E.T., L.Á.-B. and I.C.A.-G.; funding acquisition, E.T., L.Á.-B. and C.G.-G. All authors have read and agreed to the published version of the manuscript.

**Funding:** This research was funded by Severo Ochoa Program of the Government of the Principality of Asturias, grant number BP20-116 (Mrs. Celia Galve González).

**Institutional Review Board Statement:** Ethical approval was not provided for this study because the ethics committee did not exist before starting the study. The ethics committee began operating in 2019, June, and the data was collected earlier; the "Acuerdo de 25 de junio de 2019, del Consejo de Gobierno de la Universidad de Oviedo, por el que se aprueba el reglamento del Comité de Ética en la investigación" that means, "Agreement of 25 June 2019, of the Governing Council of the University of Oviedo, which approves the regulations of the Research Ethics Committee", created the ethics committee (see the public resolution at http://sede.asturias.es/bopa/2019/07/23/2019-07307.pdf (accessed on 10 November 2022) for further explanation of the process). However, this study was

carried out with consideration for the international protocols for scientific research, and in particular, in accordance with the requirements of the Declaration of Helsinki for research with human beings and Organic Law 3/2018, 5th of December, on the Protection of Personal Data and ensuring digital rights. In addition, we had the explicit permission of each participant to use their data for scientific research, with their anonymity and confidentiality assured. The participants provided their written informed consent to participate in this study.

**Informed Consent Statement:** Informed consent was obtained from all subjects involved in the study.

**Data Availability Statement:** Not applicable.

**Conflicts of Interest:** The authors declare no conflict of interest.

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
