# Peer review of "Do Virtual Campuses Provide Quality Education? A Study on the Perception of Higher Education"

_sustainability, doi:10.3390/su15010620_

Round 1

Reviewer 1 Report

The paper is interesting and well written. The  students' interest in supporting lectures and educational activities through an appropriately designed virtual environment is well known. Also, there is a necessity teachers enhance collaborative learning, focusing on the suitability/efficiency in different training scenarios, so I suggest to authors to comment on these challenges in the conclusions.

Author Response

There is a necessity teachers enhance collaborative learning, focusing on the suitability/efficiency in different training scenarios, so I suggest to authors to comment on these challenges in the conclusions: We thank the reviewer to include this suggestion in our work. For this reason, we have included in the conclusions the need to grant the virtual campus the potential to carry out activities that encourage cooperative work and learning in students, since currently the scientific literature supports the benefit that this entails for better academic performance (added in Conclusions: page 10, lines 8-13).

English language and style are fine/minor spell check required: Although the translation of this work was done by an expert translator, now the work has been revised again by another to iron out idiomatic imperfections.

Reviewer 2 Report

This is an important study since it identifies some of the limitations of virtual classes and the need for faculty members to be more proactive with students ascertaining their emotional and social needs.

Author Response

This is an important study since it identifies some of the limitations of virtual classes and the need for faculty members to be more proactive with students ascertaining their emotional and social needs: We thank this reviewer for his positive appreciation of our work.

English language and style are fine/minor spell check required: Although the translation of this work was done by an expert translator, now the work has been revised again by another to iron out idiomatic imperfections.

Reviewer 3 Report

TITLE: “Do virtual campuses grant quality to the education received? A study on the perception of Higher Education”

It is crucial interesting the use of technologies in educational contexts. However, author(s) investigated virtual campus as support for teaching. At this point, author(s) connected the quality of education with virtual campus. Therefore, I want to ask the authors how to connect the virtual campus with the quality of education. In this situation, we need to clarify the term of virtual campus and what the quality of education. Because, the authors claimed that virtual campuses support for the activities which makes them into repositories of information and content.

It is prominent thought in this paper that the universities should continue to develop, implement, and promote good practices related to virtual teaching as learning environments, technological infrastructure, teaching methodologies used, or the results of learning. I join with this thought. Therefore, I recognize that there has been a growing interest in the scientific community about the study of the quality of teaching practices via virtual campuses in higher education. Perhaps, the original aspect of this study was that there are few studies that have examined this from the student perspective.

At this point, the authors claimed that new students have different needs in terms of attention, monitoring, and evaluation of the learning process compared to more experienced students (Line 111- 112). Thus, current study was to analyze students’ perceptions, by university years as pursued, of the teaching practices delivered through the virtual campuses as a support to provide quality to their university teaching.

Regarding the Materials and Methods

The authors selected students as attending the university as first, second and third pursued years. To this point, according to my opinion, it could be selected the students in the university at first and fourth years for to be reveal significant differences between them.

At this point, additionally, it is interesting to select students in different disciplines because, there could be different expectations of the students especially in various disciplines as social and legal sciences (54.3%), engineering and architecture (3.8%), 138 health sciences (37.2%), arts and humanities (4.1%), and science (0.6%).  Therefore, this could be a handicap situation to determine students’ thoughts in virtual campus because of there is various goals of these disciplines, which can be a new research subject.

On the other hand, the authors used questionnaire that was made up of 44 items in seven dimensions, but they selected 12 items from various blocks for the current study in order to achieve their objectives, making up two dimensions of study. How to be selected these 12 items by the authors? This process was not clarified in the paper. Is this not a handicap?

These unclear points should be lighted as stated above.

Regarding the results

In this regard, authors analyzed 12 items as two Dimension as 1: The campus as a complimentary support resource adding quality to the education received and 2: The campus as a potential tool for teacher-student interaction (Comparative analysis for Dimension). Although these dimensions are not enough to assess virtual campuses, at least it can be said that presents current results in related area. However, significant differences were only found in one of the variables in these dimension, with first-year students were more in agreement that activities were published on the virtual campus which encourage the discussion of ideas, debate, etc. with a small effect size. (line 221-222).

At this time,  there could be information under the related tables in the paper about the significant differences among students as attending the university as first, second and third pursued years as related of which students in favor of non-parametric statistic technique.

The hypotheses of this study were as 1-First-year students will ascribe more importance to teachers’ teaching use of the virtual campus as a quality complement to their classes than students in later years, 2- ) first-year students will ascribe more importance to teacher-student interaction-communication driven by the teaching use of virtual campus tools, and 3- there will be no statistically significant differences between the three academic years in students’ perceptions about the teachers’ promotion of motivation in interactions via the virtual campus. Is this third hypothesis contradiction with first and second hypothesis? What is the rationale basics of these hypotheses?

As a result, - The values of the mean ranks were higher in first-year students in variables referring to communication with teachers, and -First-year students also perceived better encouragement of motivation in interactions by teachers via the virtual campus.

I have to admit that authors selected complicated subject to investigate because of there are few years between the students as attending the university as first, second and third year. It could be selected the students in the university at first and fourth year to reveal clearly significant differences between them.

In the first dimension of the analysis, authors expected first-year students to have higher ratings for teachers’ teaching practices in these spaces as supporting teaching quality: but this (H1) did not support. Continuing with the second dimension, the first-year students who perceived more encouragement of teacher-student interaction via this resource, which confirms authors’ second hypothesis (see items 7-12 in Table 3). In a shortly, the current result indicated that first-year students perceived more encouragement of teacher-student interaction but, they have no opinion for teachers’ teaching practices as supporting teaching quality via the virtual campus. Accordingly, teacher-student interaction is well; however, teachers’ teaching practices is not well level. Therefore, authors should discuss this result in terms of what causing of this result regarding educational activities and toll especially in the perspective of for teachers’ teaching practices as supporting teaching quality via the virtual campus.

The authors used a questionnaire that was made up of 44 items but selected 12 items of it to use for the current study. This situation  should be placed as a limitation  in the limitations section. In addition, the authors should state in the implications section learning outcome gained from this study.

Lastly, this study will be valuable more by the way of adding elaborated suggestions in practical ways to be implemented in virtual campus according to the current results.

Good luck,

Author Response

I want to ask the authors how to connect the virtual campus with the quality of education. In this situation, we need to clarify the term of virtual campus and what the quality of education. Because, the authors claimed that virtual campuses support for the activities which makes them into repositories of information and content: Thank you very much. We want to highlight that in the introduction, in the third paragraph, we have explained this, as indicated below: Despite that, and according to these authors, there does seem to be a consensus that virtual campuses, rather than correlating to physical university campuses, are “web sites available to an educational community, with the facility to provide teaching resources, and communication and interaction functions”. Nevertheless, although they were created to make the educational services provided by the universities more available -teaching materials, online library access, etc.- [17] nowadays, thanks to advances in educational information technology, there are many possibilities. These include better accessibility from any internet-connected device [18,19], making the teaching-learning process possible in any context and at any time [20].

It is prominent thought in this paper that the universities should continue to develop, implement, and promote good practices related to virtual teaching as learning environments, technological infrastructure, teaching methodologies used, or the results of learning. I join with this thought. Therefore, I recognize that there has been a growing interest in the scientific community about the study of the quality of teaching practices via virtual campuses in higher education. Perhaps, the original aspect of this study was that there are few studies that have examined this from the student perspective: We are pleased that this reviewer agrees on the same notion of giving appropriate importance to the fact that higher education institutions have to ensure quality education and therefore have to continue developing and improving practices related to virtual teaching.And as it is referred to, our study is relevant, because it is one of the few that give students a voice to know their appreciation of the quality of virtual teaching.

At this point, the authors claimed that new students have different needs in terms of attention, monitoring, and evaluation of the learning process compared to more experienced students (Line 111- 112). Thus, current study was to analyze students’ perceptions, by university years as pursued, of the teaching practices delivered through the virtual campuses as a support to provide quality to their university teaching: We thank this reviewer again for reading our work carefully and in detail and remembering its objective.

The authors selected students as attending the university as first, second and third pursued years. To this point, according to my opinion, it could be selected the students in the university at first and fourth years for to be reveal significant differences between them: Thanks to the reviewer for this interesting suggestion for the continuation of future work. For this reason, we have included this suggestion in the discussion section as a possible avenue for relevant future research (added at the end of the Discussion section: page 9, paragraph 4, lines 7-11).

At this point, additionally, it is interesting to select students in different disciplines because, there could be different expectations of the students especially in various disciplines as social and legal sciences (54.3%), engineering and architecture (3.8%), 138 health sciences (37.2%), arts and humanities (4.1%), and science (0.6%). Therefore, this could be a handicap situation to determine students' thoughts in virtual campus because of there are various goals of these disciplines, which can be a new research subject: This is a very interesting upcoming work suggestion. The authors have included in our work this suggestion in the possible lines of future scientific research (added at the end of the Discussion section: page 9, paragraph 4, lines 7-11).

On the other hand, the authors used questionnaire that was made up of 44 items in seven dimensions, but they selected 12 items from various blocks for the current study in order to achieve their objectives, making up two dimensions of study. How to be selected these 12 items by the authors? This process was not clarified in the paper. Is this not a handicap?: Thank you very much for this comment. We consider that we have given an explanation that is not very accurate and we have improved the instrument so that it does not give rise to errors. The 12 items were selected corresponding to the perception that the student could have about the use of the virtual campus by the professors, as well as the interaction that they could maintain through this tool, taking into consideration that the general questionnaire analyses other different aspects about campus virtual and the perception of the students use by themselves (added in page 4 in yellow).

The hypotheses of this study were as 1-First-year students will ascribe more importance to teachers’ teaching use of the virtual campus as a quality complement to their classes than students in later years, 2- ) first-year students will ascribe more importance to teacher-student interaction-communication driven by the teaching use of virtual campus tools, and 3- there will be no statistically significant differences between the three academic years in students’ perceptions about the teachers’ promotion of motivation in interactions via the virtual campus. Is this third hypothesis contradiction with first and second hypothesis? What is the rationale basics of these hypotheses?: The reviewer's comment is greatly appreciated. We would like to clarify this issue. The first hypothesis refers to the first dimension, which mentions that first-year students will give more importance to the use of the virtual campus by the teacher, as a quality complement to their classes. The second hypothesis refers to the second dimension: first-year students will give more importance to the interaction between teachers and students through the campus than those in later courses. However, the third hypothesis refers to one of the more specific items in the questionnaire, considering that the promotion of motivation through the campus is important for all of them, that is, it is not associated with a specific course.

I have to admit that authors selected complicated subject to investigate because of there are few years between the students as attending the university as first, second and third year. It could be selected the students in the university at first and fourth year to reveal clearly significant differences between them: Indeed, it would be very interesting to study whether there are high statistically significant differences between the perception of students who have recently arrived at the university and those who are about to leave the educational institution. For this reason, this relevant suggestion has been commented on in our modified work in the possible future lines.

In the first dimension of the analysis, authors expected first-year students to have higher ratings for teachers’ teaching practices in these spaces as supporting teaching quality: but this (H1) did not support. Continuing with the second dimension, the first-year students who perceived more encouragement of teacher-student interaction via this resource, which confirms authors’ second hypothesis (see items 7-12 in Table 3). In a shortly, the current result indicated that first-year students perceived more encouragement of teacher-student interaction but, they have no opinion for teachers’ teaching practices as supporting teaching quality via the virtual campus. Accordingly, teacher-student interaction is well; however, teachers’ teaching practices is not well level. Therefore, authors should discuss this result in terms of what causing of this result regarding educational activities and toll especially in the perspective of for teachers’ teaching practices as supporting teaching quality via the virtual campus: We thank the reviewer for reminding us of some of our results. Likewise, we take note of the relevance of further discussing them in our work (added in the Conclusions section, page 10, lines 13-15).

The authors used a questionnaire that was made up of 44 items but selected 12 items of it to use for the current study. This situation should be placed as a limitation in the limitations section. In addition, the authors should state in the implications section learning outcome gained from this study: We are very grateful to the reviewer for this interesting suggestion. However, the authors have not included it in the manuscript since we do not consider it as a limitation. The initial questionnaire prepared by the authors, although it is true that it consists of 44 items grouped into 6 dimensions; for this specific work, the objective was not to use all the dimensions. For this reason and based on the hypotheses of this new work, 12 items were selected from all the dimensions of the initial questionnaire to assess the quality of virtual teaching, assessing it from two perspectives: 1. teaching practice on the virtual campus as support the quality of teaching and 2. the use of the campus to improve interaction with students.

Lastly, this study will be valuable more by the way of adding elaborated suggestions in practical ways to be implemented in virtual campus according to the current results: Thank you very much for this interesting and important suggestion. Faithful to it, we have put several actions to improve teaching practices in online teaching (added at the end of the last paragraph of the Conclusions section in yellow).

Round 2

Reviewer 3 Report

Dear Authors,

I have read your responses about the revision study (Manuscript IDsustainability-2065609). I have seen that you added important texts to the paper that I indicated points in the paper.

Congratulations.